# Glucagon-like Peptide-1 (GLP-1) Receptor Agonists and Cancer Prevention: Methodological Pitfalls in Observational Studies

**DOI:** 10.3390/cancers17091451

**Published:** 2025-04-26

**Authors:** Matthew Harris, Michelle Harvie, Andrew G. Renehan

**Affiliations:** Division of Cancer Sciences, School of Medical Sciences, Faculty of Biology, Medicine and Health, University of Manchester, Manchester M23 9LT, UK; michelle.harvie@manchester.ac.uk (M.H.); andrew.renehan@manchester.ac.uk (A.G.R.)

**Keywords:** obesity, cancer, glucagon-like peptide-1 (GLP-1) receptor agonists

## Abstract

Obesity is linked with an increased risk of at least 13 cancer types. However, there is limited, low-quality evidence to show that losing weight in individuals with obesity results in reduced cancer risk. Glucagon-like peptide-1 (GLP-1) receptor agonists are a class of anti-diabetes drugs with proven efficiency in the treatment of diabetes, which have recently shown striking results as weight loss medications, such that it is conceivable that long term use might be useful to prevent cancers associated with obesity. Currently, however, GLP-1 trials are not large enough nor follow-up long enough to answer this question, so researchers use ‘real world’ data or observational studies to look at this question. This opinion paper highlights that there are many pitfalls with this approach and several methodological aspects to analysing the data, all of which need to be considered.

## 1. Introduction

In 2016, the International Agency for Research on Cancer (IARC) convened an international working group to review the large body of evidence linking obesity, commonly approximated by body mass index (BMI), with cancer incidence. Using a framework of evidence evaluation, the group concluded there was sufficient strength of evidence to causally link excess body fatness with 13 cancer types [1]. In reaching the conclusion of causality, an association had to be observed in studies in which chance, bias, and confounding could be ruled out with confidence, and in which the human data were supported by equivalent observations in animal studies, and there was mechanistic plausibility. The thirteen cancers were oesophageal adenocarcinoma; cancers of the gastric cardia, colon and rectum, liver, gallbladder, pancreas, post-menopausal breast, corpus uteri, ovary, kidney (renal cell), thyroid; multiple myeloma; and meningioma. These are collectively referred to as obesity-related cancers [2]. More recently, two large population-based studies [3,4] and an updated meta-analysis [5] suggest that the number of obesity-related cancers might be closer to twenty, but these additional cancer types have not been evaluated against IARC criteria for causality.

Against this background, and the global public health concern of the obesity epidemic [6], there is a need to consider strategies to reduce the risk of obesity-related cancers through weight loss interventions [7]. To date, there are no trials of a weight loss intervention with cancer as the primary outcome. In the Look AHEAD study [2], a trial designed to test an intensive lifestyle and weight loss intervention (5 years) to reduce cardiovascular mortality in patients with type 2 diabetes, a tertiary analysis found that, at a median 11 years after the initial intervention, there was a 16% reduction in the risk of obesity-related cancers, but this did not reach statistical significance. Data from observational studies [8] where weight loss has been recorded have been inconsistent as it is near impossible to disentangle intentional from non-intentional weight loss. Bariatric surgery is associated with a sustained weight loss in the order of 20–30% at 10 years. Data from large-scale studies [9] and meta-analyses [10] indicate risk reductions in several obesity-related cancers. It is hypothesised that this cancer risk reduction is mediated through weight loss, but other mechanisms (e.g., reduced weight-independent insulin resistance) [11] might also have a role.

Into this arena has now entered promising pharmacological interventions for weight loss. The glucagon-like peptide-1 (GLP-1) receptor agonists are a class of incretins originally designed for use as anti-diabetes medications, which now have a proven track record for weight loss. Semaglutide, when administered in doses higher than those used in the management of diabetes, is associated with approximately 15% weight loss in individuals with obesity without diabetes [12], and accordingly received a licence in the USA for this indication in 2021. These data are based on mainly short-term trials, with the largest trial, SELECT [13], reporting data with 34 months of drug or placebo use. Semaglutide reduces major disease endpoints, such as deaths from kidney disease [14] and cardiovascular disease [13], among obese individuals without disease. It is generally felt that, in the main, the beneficial effects of semaglutide are mediated through its weight loss effect [14]. The impressive, sustained weight loss with these medications leads to the hypothesis that these drugs might have a role in the prevention of obesity-related cancers, mediated through weight loss.

In the absence of long-term follow-up and large-scale trial data, researchers turn to observational studies to address the above hypothesis. One such example was recently reported in this journal by Levy and colleagues [15]. The present article aims to critically appraise the methodological challenges and pitfalls associated with studying weight loss and cancer risk reduction in observational studies, and to exemplify this through critique of the Levy paper.

## 2. Methods

To address these methodological challenges, we modified the ROBINS-E framework for assessing risk of bias [16] and supplemented this with reference to previous work, under the auspices of the European Association for the Study of Diabetes (EASD) Diabetes and Cancer Research Consortium (DCRC), with which one of the present authors (AGR) was involved [17,18,19]. Specifically, the latter was related to the use of anti-diabetes medications and cancer risk [20]. The DCRC group, and others, ultimately demonstrated that the first generation of observational studies had design flaws such that when these design issues were addressed, in the main, associations of metformin [21] and insulin with cancer disappeared.

There are seven methodological criteria specific to this research question against which data should be interpreted:

### 2.1. Adequate Adjustment for Key Parameters of Body Fatness

Lifetime exposure to excess body fatness (commonly approximated by BMI) is associated with increased risk of obesity-related cancers and is also an indicator for initiation of treatment with GLP-1 agonists—thus the risk for confounding. This can be addressed through adjustments in regression models, stratification methods or through matching.

### 2.2. Immortal Time Bias

We have previously defined this as a statistical pitfall that “is conceptually difficult and perhaps not well appreciated by many researchers… Individuals have to survive to the start of exposure—hence, the name of ‘immortal’. The result is an advantage to the users of the drug of interest when the analysis is simply categorised by ever/never use” [20]. This problem was exemplified by Suissa et al. [21] demonstrating that immortal time bias was prevalent in many observational studies reporting reduced cancer risk with metformin use, while studies that used methods to avoid these biases reported no effect of metformin use on cancer incidence.

### 2.3. Treatment Allocation/Selection Bias

This arises where there is a non-random allocation of a drug due to variation in clinical practice and judgements made by clinicians. In turn, these are based on various factors including patient characteristics, predicted adverse effects, access to the drug, and healthcare or health insurer policies. This is particularly prevalent in pharmaco-epidemiological studies and is sometimes referred to as *confounding by indication*. An example is when insulin therapy is prescribed late in the course of diabetes when patients are older and ‘sicker’ [18]. These biases tend to result in an apparent increased cancer risk. Methodological solutions require advanced methods such as inverse probability weighting and modelling the data with time-updated terms in the model.

### 2.4. Survival Bias

This bias is due to differential survival (or differential censoring in follow-up) in patients receiving the drug of interest compared with those in the comparator group. In turn, this will differentially influence the number of individuals at risk to develop the event of interest, namely incident cancer. For example, among individuals with obesity without diabetes and receiving GLP-1 agonists, there is an approximately 20% risk reduction in cardiovascular mortality [13]; thus, in those individuals taking these drugs, more may grow older and present with an apparent increase in cancer risk. The methodological solution here is to use competing risk analyses [22].

### 2.5. Cumulative Drug Dose Effect

In general terms, a cumulative drug dose effect on the outcome, i.e., a higher cumulative drug dose is associated with greater risk reduction, supports a causal association. The DCRC group previously argued that analyses of the cumulative effect of a drug on outcome in models that also include time-dependent covariate adjustment for ever vs. never use is one way to avoid such between-person allocation bias [20].

### 2.6. Sufficient Sojourn Time Between Drug Intervention and Cancer Presentation

It is generally accepted that the duration of exposure of excess body fatness and increased cancer risk is of the order of a decade [23]. This might vary between cancer types, but the sojourn time is unlikely to be measured in a small number of years. Similarly, it is reasonable to extrapolate that the duration of drug administration required to see an effect on cancer prevention is likely to be a decade. This is illustrated for example, in the SOS study of bariatric surgery and breast cancer risk, where there is a clear separation of the cancer risk curves after 10 years [24,25]. Furthermore, if this effect is mediated via weight loss, we know that cessation of GLP-1 agonist use is quickly associated with weight regain toward the baseline weight status [26].

### 2.7. Treatment Effect Specific to Obesity-Related Cancers

It is reasonable to assume that if the effect of GLP-1 agonists is through weight loss, the treatment effect will be limited and specific to obesity-related cancers (for example, in the Look AHEAD trial of lifestyle and weight loss intervention, the non-significant risk reduction was limited to obesity-related cancers but no effect for non-obesity-related cancers [2]. Non-obesity-related cancers are unlikely to benefit from GLP-1 receptor agonist administration unless there is a weight-independent effect.

## 3. Results

Levy and colleagues [15] addressed the question of cancer risk reduction with GLP-1 receptor agonist use in adult patients with obesity in a large-scale cohort study using the TriNetX US Collaborative Network database (2013–2023). The researchers used propensity scores to match patients who received GLP-1 agonists with controls and followed them until cancer over 5 years. Their analysis found significant cancer risk reductions associated with GLP-1 agonist use in several cancer types compared with matched controls. There were reductions in the following cancer groups: gastrointestinal (HR 0.67, 95% CI 0.59–0.75), skin (HR 0.62, 95% CI 0.55–0.70), breast (HR 0.72, 95% CI 0.64–0.82), female genital (HR 0.61, 95% CI 0.53–0.71), prostate (HR 0.68, 95% CI 0.58–0.80), and lymphoid/haematopoietic cancers (HR 0.69, 95% CI 0.60–0.80). The protective effects were greater for semaglutide compared with liraglutide. The authors concluded that their study “provides compelling evidence for GLP-1 agonist’s potential role in cancer-risk reduction, with semaglutide showing particularly promising results”.

In Table 1, we have listed the seven methodological criteria specific to the index research question and assessed for risk of bias. Six of the seven assessments revealed high risk of bias in the Levy study.

## 4. Conclusions

There is biological plausibility that weight loss could reverse an obese individual’s risk of cancer. However, there has not been conclusive evidence to show this link is generalisable across all obesity management interventions, and whether the causal effect is due to weight loss alone. Bariatric studies have shown the potential for considerable cancer risk reductions [10], with behavioural interventions suggesting more modest weight loss, but still present protective effects [2,8]. GLP-1 agonist therapies present an intermediate weight loss option, which may then present a cancer risk reduction between these points. Issues with availability [27], adherence [28], side effects [29], and weight regain [30] must be considered in the real-world efficacy.

A spectrum of challenges to observational data analysis are posed in the effort to understand the potential cancer-prevention effects of GLP-1 receptor agonists. Various solutions for future analytical plans have been discussed. On the surface, the findings from the Levy et al. [15] study look promising. However, we believe that further research, applying the study design framework outlined here, is needed before evidence will be strong enough to influence clinical management and public health policy and to consider incretin drugs as a potential cancer risk-reducing medication strategy. This may take the form of mediation analysis to disentangle the pathways of cancer risk reduction in observational studies of this nature or a large-scale clinical trial, if feasible. It is also key to understand the causal pathways for these drugs and their potential in cancer prevention, whether it is via weight loss (indirect effect) or a direct anti-cancer effect of the GLP-1 drugs themselves (direct effect).

## Figures and Tables

**Table 1 cancers-17-01451-t001:** Modified risk of bias in non-randomised studies of exposure (ROBINS-E) relevant to the Levy et al. [15] study.

Domains	ROBINS-E Criteria	Modified ROBINS-E Specific to the Question of GPL-1 Agonist Exposure	RoBRed = high Orange = IntermediateGreen = Low	Comments
1	Risk of bias due to confounding	Adequate adjustment for key parameters of body fatness, such as BMI	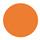	The study accounted for the potential confounding of BMI and was matched for this through propensity scoring. However, the final models do not adjust for BMI, so there may still be residual confounding.
2	Risk of bias arising from measurement of exposure	Immortal time bias	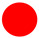	For pharmaco-epidemiology studies, this requires that only new users of the drug are considered rather than ever/never users, as applied in this study. There are various statistical modelling approaches (e.g., time-varying Cox model) that can be used to address this. The Levy paper did not include a methodology such as this.
3	Risk of bias in selection of participants for study (or for analysis)	Treatment allocation/selection bias	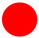	This is a universal problem in all non-randomised studies. This bias can be reduced through various techniques such as inverse probability weighting, and by including the treatment allocation in the model.The study period was from 2013 to 2023; yet, semaglutide was only licenced for management of obesity with diabetes from 2021 in the USA. Semaglutide was not licenced for use in diabetes until 2017.Many patients in this study had diabetes as well as obesity. This was not a ‘obesity without diabetes’ study similar to the STEP series of trials and the SELECT trial.
4	Risk of bias due to post-exposure interventions	Survival bias	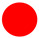	To address this bias, there is a need to consider the competing risk of non-cancer deaths. For example, patients on long-term semaglutide have a 20% reduction in CVD mortality.
5	Risk of bias due to missing data	Cumulative drug dose effect	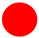	In general terms, a cumulative drug dose effect on the outcome, i.e., higher cumulative drug dose = greater risk reduction, supports a causal association. The drug doses are missing in this study. Additionally, in this study, it is unclear whether the high dose of semaglutide (2.4 mg) is the exposure of interest.
6	Risk of bias arising from measurement of outcome	Sufficient sojourn time between drug intervention and cancer presentation	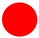	It is generally held that the sojourn time from obesity exposure to increased cancer incidence is in the order of a decade. It is reasonable to assume that a similar period would be required to see the effects of risk reduction from a drug. Yet, in this study, there were substantial risk reductions within a year. This does not seem biologically plausible.
7	Risk of bias in selection of reported result	Treatment effect specific to obesity-related cancers	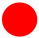	The hypothesis being tested is that GLP-1 agonists protect against cancer mediated through weight loss. To test this, the outcome of interest is obesity-related cancers. Taking this further, mediation analysis will help to disentangle cancer protective effects due to weight loss and direct drug action.

GLP-1: glucagon-like peptide-1. BMI: body mass index. RoB: risk of bias. CVD: cardiovascular disease.

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
