# Peer review of "Glucagon-like Peptide-1 (GLP-1) Receptor Agonists and Cancer Prevention: Methodological Pitfalls in Observational Studies"

_cancers, 2025, doi:10.3390/cancers17091451_

Round 1
Reviewer 1 Report
Comments and Suggestions for Authors
This article critically evaluates methodological challenges when studying GLP-1receptor agonists for cancer prevention via weight loss. It highlights promising findings from observational studies suggesting GLP-1 agonists, particularly semaglutide, might reduce cancer risk. However, the authors identify substantial methodological pitfalls including: immortal time bias, inadequate control for confounding factors like BMI, selection and survival bias, lack of cumulative drug dose and insufficient sojourn time. They conclude that current evidence, while suggestive, remains limited due to high risk of bias, underscoring the need for rigorous, well-designed studies. It is worth publishing the article and it would benefit from improving the methods section.
Author Response
Thank you for your comments. The reviewer comments and replies are attached in the word document.

Reviewer 2 Report
Comments and Suggestions for Authors
The authors evaluated Levy et al.'s study and analyzed it, revealing the following issues: inadequate adjustment for confounding factors (such as BMI), time to immortality bias, treatment allocation bias, survivor bias, lack of cumulative drug dose effect, and unreasonable latency period. Finally, it is emphasized that the existing evidence is not sufficient to support GLP-1 agonists as a cancer prevention strategy, and further mediation analysis and clinical trial validation are needed. But there are some issues with the article:
- The keyword "GLP-1" should ideally be the full name "Glucagon like peptide-1" plus the abbreviation "GLP-1";
- Why does “Objectives” appear in “2.Materials and Methods”? Should the content of “Objectives” be placed at the end of “introduction”?
- Is the abbreviation for "Glucagon like peptide-1" GLP-1 "or" GPL-1 "? Please maintain consistency;
- After the abbreviation "GLP-1" appears once in the main text, it can be abbreviated directly afterwards;
- After "3.Results" is "5.Conclusion", should there be "4.Discussion" in between?
- Inconsistent reference format.
Author Response

(The authors gave the same response as above.)

Reviewer 3 Report
Comments and Suggestions for Authors
This is an interesting review where the authors applaud the potential of GLP-1 agonists to be used in cancers. They basically demonstrated certain downsides and loopholes that are linked to studying whether weight loss with GLP-1receptor agonists reduces cancer risk in observational studies, and exemplifies this through critique of a recent article published in the journal.
- It's clear that not enough studies are available to claim the GLP-1 as robust anticancer agents; nevertheless, there needs to be a small discussion on their foreseeable efficacy in cancers.
- Weight loss could be beneficial but it's kind of ambiguous whether only weight loss is practically associated with anticancer effects. Hence, addition of another small discussion/paragraph could be a good idea to show how it is linked to anticancer effects.
Author Response

(The authors gave the same response as above.)
